# Socialization, Relaxation, and Leisure across the Day by Social Determinants of Health: Results from the American Time Use Survey, 2014–2016

**DOI:** 10.3390/healthcare11111581

**Published:** 2023-05-28

**Authors:** James Davis, Deborah A. Taira, Eunjung Lim, John Chen

**Affiliations:** 1Department of Quantitative Health Sciences, John A. Burns School of Medicine, 651 Ilalo Street, Honolulu, HI 96813, USA; lime@hawaii.edu (E.L.); jjchen@hawaii.edu (J.C.); 2Daniel K. Inouye College of Pharmacy, 722 South Aohoku Place, Hilo, HI 96720, USA; dtjuarez@hawaii.edu

**Keywords:** social determinants of health, activity, sports, sleep, ethnicity

## Abstract

This study used the American Time Use Survey (ATUS) to examine socialization, relaxation, and leisure activities throughout the day as related to social determinants of health (SDOH). The study population was adults aged 25 years and older who participated in the ATUS in 2014–2016, the most recent years for collecting SDOH. Descriptive analyses provide characteristics of the study population. Graphical analyses display socialization by SDOH across the hours of the day based on adjusted regression models. Quasi-binomial models analyzed the association between the numbers of minutes of various activities and SDOH. Associations between SDOH and sleeplessness (yes or no) were explored using logistic regression. For much of the day, being female, having less education, living in poverty, and having food insecurity were associated with more time socializing and relaxing. The major activities under socializing and relaxation are watching television and movies. Having a college degree was strongly associated with increased minutes of sports activity, whereas living in poverty and food insecurity were associated with fewer minutes. Less education, living in poverty, and having food insecurity were associated with sleeplessness. A possible mechanism of the effects of SODH on health is by its altering of the patterns of daily life.

## 1. Introduction

Daily activities, including the amount of socialization, relaxation, and leisure time, can impact physical, mental, and emotional well-being. Social interaction, the strengthening of friendships, and other social activities can enhance mental health, lessen stress, and provide a sense of community and support [1]. Relaxation techniques, including deep breathing, meditation, or engaging in enjoyable leisure activities, can lessen stress, anxiety, sadness, and other mental illnesses [2]. Numerous health problems, including heart disease, a weaker immune system, anxiety, and depression, have been linked to high levels of stress [3].

Socialization, relaxation, and leisure activities can impact health through several pathways, including amount of sleep and its quality, and physical activity. Both positive and negative impacts on sleep might result from interacting with others and participating in social activities. Spending quality time with loved ones can foster happiness, connection, and well-being, which can lead to better sleep [4]. Stress and anxiety can be lessened by social support and a sense of belonging, which can make it simpler to get to sleep and stay asleep [5]. However, socialization might lead to arguments, peer pressure, or having too many social obligations that might cause anxiety, making it difficult to unwind and go to sleep [6]. Moreover, sleep quality can be directly impacted by relaxing activities. Deep breathing, meditation, progressive muscle relaxation, and listening to calming music are relaxation techniques that can help calm the mind and body and get them ready for sleep by lowering stress, anxiety, and racing thoughts [7]. Engaging in enjoyable leisure activities can also have a positive impact on sleep. Hobbies, sports, or creative leisure endeavors may improve mood and lead to relaxation that can lead to better sleep. However, the frequency and duration of leisure activities matters. It could be harder to get to sleep if you engage in stimulating activities too soon before bed [8]. In addition, leisure activities offer a chance for physical and mental renewal, which can improve general wellbeing and athletic performance [9]. Athletes can find a balance and avoid the monotony or burnout brought on by rigorous training programs by engaging in pleasurable activities such as hobbies, spending time in nature, exploring creative outlets, or doing other enjoyable things; however, although other leisure activities, such as watching movies or television shows, can be a relaxing, spending too much time on screens can lead to reduced physical activity that can lead to weight gain, muscular deterioration, chronic health conditions, and mental health issues [10]. Hence, how a person spends their day and the decisions they make regarding socialization, relaxation, and leisure can greatly impact their health status.

The World Health Organization defines social determinants of health (SDOH) as the conditions in which people are born, grow, live, work and age that affect a wide range of health risks and outcomes [11]. Given this broad definition, it is difficult to assemble a complete list of SDOH; however, education, living in poverty, and food insecurity have been shown to relate to worse health across a wide spectrum of diseases [12,13,14,15]. The World Health Organization (WHO) has proposed a theoretical framework that recognizes that SDOH are complex and multifaceted, leading to health outcomes through multiple pathways [16]. The framework includes three main components: structural determinants, the social, economic, and political conditions that drive inequalities; intermediary determinants that mediate the structural determinants, and health outcomes. We theorize that SODH can create a pathway to health outcomes through differences in day-to-day behaviors. Specifically, we hypothesize that SODH will have measurable short-term effects such as decreased physical activity during the day and sleep problems at night, and potential long-term health effects through chronic stress and sustained inactivity.

The objective of this study is to examine how social determinants of health (e.g., education, poverty, and food insecurity) affect time spent on various activities, including socialization, relaxation, and leisure, utilizing the American Time Use Survey (ATUS) data [17,18]. The ATUS interviews a randomly selected subset of the Current Population Survey and asks detailed questions about how they spent their day. The findings from this study will provide a better understanding of how the daily lives of Americans vary depending upon SDOH. 

## 2. Methods

### 2.1. Study Design

The study was designed to understand how the daily lives of Americans vary depending upon social determinants of health. The study used the ATUS, a long-term survey of activities done during a day, conducted annually since 2003. The analysis used de-identified, publicly available data, and therefore does not constitute human subjects research as defined at 45 CFR 46.102. Patient consent was waived due to this exemption. Households in the ATUS are a random subset from households completing their eighth month of interviews for the Current Population Survey, the nation’s monthly labor force survey. A randomly selected individual aged 15 years or older is invited to participate in the ATUS. 

### 2.2. Study Population

The ATUS sometimes adds additional modules on topics of national concern, and our study population includes respondents that participated in the Eating and Health Module (EH), which is collected in selected years. The EH is a component of the ATUS that includes information on SDOH, the focus of our study, in addition to the data regularly collected on activities during the day. The latest years including the EH module were 2014–2016, the source of data for our study. A total of 32,990 people were interviewed in these years, with response rates of 46.8–51.0%. Participants in the analysis were selected to be White, Asian, or Black, and of ages 25 years or older. We restricted the population to people aged 25 and older so that they would have time to complete their highest educational attainment. Participants were excluded if the interviewer perceived their data to be of poor quality, if survey weights were not positive, and if their data did not include the complete 1440 min (24 h)-worth of data possible in a day. The ATUS defines poor-quality data as those containing fewer than five activities, those for which refusals or “don’t remember” responses account for 3 or more hours of the 24 h day, or both. The final sample size for the analysis was 20,737 participants. 

ATUS survey administration: Households in the ATUS are a random subset from households completing their eighth month of interviews for the Current Population Survey, the nation’s monthly labor force survey. A randomly selected individual aged 15 years or older is invited to participate in the ATUS. The sample is randomized by day with half of the participants asked about a weekday and half about a weekend day. For the selected day, respondents are asked about how they spent their time, where they were and whom they were with. The ATUS is conducted using computer-assisted telephone interviewing. The interview uses a flexible interviewing technique to help make reporting on activities comfortable and accurate. The technique allows interviewers to guide respondents for details required to code activities and avoid unnecessary information. After the interview is completed, coders assign codes to the reported activities. The tasks of processing, editing, and coding the data are expedited because all of the collected information is stored in the computer.

Poverty was defined as being 185% of the poverty threshold. These thresholds are the income-eligibility thresholds for food and nutrition assistance programs. The ATUS used fiscal year 2014 poverty thresholds for 2014, and fiscal year 2016 poverty thresholds for 2015 and 2016. Food insecurity was taken as participants answering that they sometimes or often did not have enough to eat.

### 2.3. Study Outcomes

A primary outcome is the total minutes spent in socializing, relaxing, and leisure, a category defined by the ATUS. Major activities within this activity include television and movies, socializing and communicating with others, reading for personal interest, relaxing and thinking, playing games, and computer use for leisure excluding games. Other outcomes are the total minutes of moderate and vigorous activity, minutes of sports activity, and minutes of sleeplessness. Sleeplessness was qualitative in the sense that participants were coded as reporting an interval of sleepless (yes or no).

### 2.4. Statistical Analysis

The ATUS often is not considered a longitudinal database, since people are only interviewed once, but their activities are monitored longitudinally for the full 1440 min in a day. Our study takes advantage of this detailed data collection by calculating the numbers of minutes spent doing activities throughout the day. Our study is a longitudinal look at behavior in lives across a single day. For the descriptive and regression analyses, minutes of activities were summarized by the hours of the day. Analyses were limited to the hours from 5 a.m. through midnight, except for sleeplessness. Few people performed the studied activities from midnight to 5 a.m.

Individuals with poor-quality data were excluded from analyses. Poor quality was defined by a field in the database of the ATUS. Questions of poor quality were flagged, and we excluded individuals with poor-quality data from our study. 

Initial analyses were descriptive, providing the number of people by the characteristics of the study population. For these descriptive analyses and subsequent regression models, ATUS weights were used to account for the survey design of the EH module. Because the ATUS is a random sample from the Current Population Survey, a nationally representative sample of the United States population, the analyses using complex survey weights maintained the representative interpretation.

Quasi-binomial regressions were used to model the minutes of socializing, relaxing, and leisure. The relative counts with 95% confidence intervals (CIs) are reported. Sleeplessness was binary, and reporting of an interval of sleeplessness of any length was taken as evidence of sleeplessness. Sleeplessness in the previous day was infrequent, making logistic regression a reasonable choice. Results are provided as odds ratio with 95% CIs. Moderate and vigorous activity was defined as activities of 3 metabolic equivalents (MET) or higher, i.e., a ratio of your working metabolic rate relative to your resting metabolic rate of 3 or higher. Since fewer people had answered the poverty and food insecurity questions, to test the robustness of parameter estimates, we conducted two models—Model 1 with all the covariates without poverty and food insecurity, and Model 2 with all the covariates including them.

To allow flexibility in the shapes of the associations in the figures, separate regression models were fit by the hour of the day for the various outcomes. The figures are limited to weekdays to show activity during a typical weekday. The figures present results drawn with 95% confidence bands. Some activities comparing weekdays and weekday are summarized as odds ratio of the minutes with activities and presented with 95% CIs. 

The data are freely available with consent at https://www.atusdata.org (accessed on 18 May 2021). The ATUS data were extracted using the IPUMS data extraction tool [19]. Analyses were performed using R version 4.1.1 with analyses of complex survey data undertaken using the survey package (v4.1.1; Lumley, 2021) [20].

## 3. Results

The largest age group was participants ages 25–49 years old (46.4%); 30.0% and 23.6% were ages 50–64 years and ages 65 years and older, respectively (Table 1). The proportions were close to equal by sex and about equal by design in reporting activities on a weekday or a weekend. About 90% were White, with 14.1% Black and 5.1% Asian. Only 6.5% had not attained an educational degree; 55.7% had a high school degree, and 37.8% a college degree. Based on reported income, 29.3% were living in poverty, and 4.4% reported food insecurity.

Most respondents spent time during the previous day socializing, relaxing, and in leisure (95.6%), with a median among those who had of 155 min (25th percentile = 285, 75th percentile = 463). A majority reported having engaged in moderate or vigorous activities (59.5%), with a median (25%, 75% percentiles) of 105 (45, 215) min. Sports activities were reported by 19.1% of the participants; the median (25%, 75%) percentiles for these participants were 60 (40, 120) min.

It is worth emphasizing that the information on activities such as sports participants and sleeplessness was not collected by survey questions. The time use data were collected by conversational interviewing rather than by scripted questions. Experienced interviewers obtained a census of activities across the 24 h (1440 min) of the previous day, activities that may or may not include sports participation and sleeplessness. Activities reported were secondarily coded into categories. The duration of activities is known to the minute, unlike surveys that more typically elicit an average effect. The time use data are valuable to the US government for learning about time spent on paid and unpaid activities (e.g., childcare, eldercare); learning the number of hours people work at home and other locations; and understanding how a particular policy might affect people’s behavior. 

The figures illustrate a typical weekday. On weekends people spent 32% more time socializing, relaxing and in leisure (95% 95% CI = 29–32%), 10% more time in moderate/vigorous activities (95% CI = 5–14%), and 42% more time (95% CI= 31.5–55%) doing sports activities. We chose to focus on socialization during weekdays, when we expect fewer people to be socializing. Socializing on weekends may have a different interpretation. 

Figure 1 illustrates the patterns of socializing including time spent relaxing and in leisure provide by age, sex, race, and education.

Older ages spent more time across the day than younger ages. People aged 65 years and older peak in the morning, with about a 75% increase relative to those aged 25–49 years, which diminishes throughout the day. People aged 50–54 years also spent more time socializing, relaxing, and in leisure than the youngest age group, with a morning peak that leveled off in the early afternoon. Women spent less time socializing, relaxing, and in leisure than men, but went from about 30% less at 6 a.m. to closer to 10% less by the early afternoon. Time spent with children may have limited the time for socializing, relaxing, and leisure. As an example, from 8 to 11 a.m., women spent 44% more time with children then men (95% CI = 29–62%), but from 5–8 p.m., the difference was only 14% (95% CI = 6–23%). Blacks contrast with Asians and Whites in spending more time in socializing, relaxing, and leisure activities across the day, but the three races became more similar in the evening hours. Educational differences were greatest from about 9 a.m. until about 6 p.m.—the center of the day—converging in the early evening. Across most hours, the college-educated participants spent the least time, and people with less than a high school degree the most. People living in poverty had increased socialization, relaxation, and leisure during the early morning, until about 9 a.m.; then, minutes socializing decreased across the day and early evening (Figure 2).

People lacking food security had a similar pattern: the odds ratio in minutes socializing peaked at about 9 a.m., followed by a steady decline. Both poverty and food insecurity were associated with 20–30% increased minutes of socializing in the morning.

### 3.1. Moderate and Vigorous Activity and Sports Activity

The total time spent in moderate and vigorous activities decreased with increasing age, decreasing to about a fourth by ages 65 years and older relative to ages 25–49 years (Table 2). Sports activities showed a similar decline with age. 

Women were less active than men, having 29% fewer minutes of moderate and vigorous activity and 41% fewer minutes of doing sports activities. Compared to Whites, Blacks had about 40% less moderate to vigorous activity and a similar difference in sports activities. Asians presented a different pattern: Asians had 24% fewer minutes of moderate and vigorous activities than Whites, but similar minutes of sports activities. By education, participants with a high school degree and those without a degree did not show a statistically significant difference. People with a college degree had 15% fewer minutes of moderate and vigorous activity, despite 60% more minutes of sports activity. 

Both living in poverty and having enough food were negatively associated with minutes of moderate and vigorous activity. The relative minutes of sports activity were even lower when comparing people with to those without food insecurity. 

### 3.2. Sleeplessness

Sleeplessness was moderately higher in women than in men, and in older ages relative to ages 25–49 years (Table 3). Blacks and Whites were similar, whereas Asians reported half the prevalence of sleeplessness of Whites. 

Sleeplessness decreased with increasing education. People without a college degree were 34% more likely to report sleeplessness than people with a high school degree, whereas those with a college degree were a third less likely to report sleeplessness than those with a high school degree.

People living in poverty had an estimated 21% more minutes of sleeplessness than people not living in poverty. Increased minutes of sleeplessness were especially high for people with food insecurity relative to those without (an estimated 44% higher).

## 4. Discussion

Our aim in this article was to explore how the daily lives of individuals differ by SODH. We hypothesized that SODH may modify lives in ways that have measurable short-term effects, such as decreased physical activity and sleep problems, and potential long-term health effects from sustained stress and inactivity. While health disparities related to SDOH are well documented, little is known with respect to how SDOH relate to activity patterns throughout the day. Our results show consistent differences in time spent during the day by age, sex, race, education, poverty, and food insecurity. Those with SODH spend more time in socialization, relaxation, and leisure activities, such as watching TV. They undertake less moderate and vigorous activity, and more often experience sleep problems. 

From a public health perspective, improving SDOH is a keystone of the Healthy People 2030 objectives. The objectives target economic stability, education, and social context—areas our study addresses. Our study of a representative national sample of the US (n > 20,000) revealed striking differences in activity patterns related to demographic characteristics and SDOH. Minutes spent socializing, relaxing, and in leisure were related to increased age, being male, having less education, being in poverty and having food insecurity. We found that Blacks and Asians spent less time in moderate or vigorous activity than Whites, and Blacks spend less time in sports activity than Whites. Compared to those with a high school degree, the college-educated spent less time in moderate or vigorous activity but more time in sports activities. 

Another activity monitored in our study was sleeplessness. Sleeplessness was positively associated with being female and being 65 years and older, and negatively associated with increasing education level and being Asian, relative to being White. Prior studies have found sleeplessness to be associated with low income and increased age [21]. Obesity, which is also strongly associated with SDOH, has been shown to affect sleep duration and quality [22]. Moreover, low neighborhood socioeconomic status and worsening neighborhood conditions were associated with unhealthy sleep behaviors [23,24,25]. Our findings related to race/ethnicity are important because many studies do not disaggregate findings from sleep apnea or insomnia by race/ethnicity [26]. 

Socialization has many positive benefits for mental health, and spending a lot of time socializing might mean one is spending less time in other activities that are also good for health, such as sports or other moderate- to high-level activities. Further research is needed to better understand the relationship between SES and socializing, as it may differ among cultures and contexts, and that there might be other factors, such as social norms, social capital, and access to social resources, that mediate this relationship [27,28].

There are several limitations to our study. First, while the ATUS provides detailed data on activity levels, it does not include information on health outcomes that can be linked to these data. Second, the small number of individuals from racial/ethnic groups other than Whites, Blacks, and Asians made it impossible to categorize these other groups separately. Third, ATUS captures only a day of activity for each individual, so we cannot examine activity levels over longer time periods, such as how much time someone spends in sports activities in a week. Forth, the study could also be affected by selection bias, recall bias, and confounding. Only about half of the eligible participants from the Current Population Study joined the ATUS. The participation may not be fully representative of the original national sample. Interview data from the ATUS asking about the previous day’s activity could incur recall bias and potentially lead to non-differential or differential misclassification. Our multivariable regression models adjusted for the major variables of interest to our study, but confounding from other variables could occur. We used data from 2014–2016, the most recent data from the ATUS with the social determinants of health we investigated. Our findings are consistent across multiple social determinants, and suggest that differences in daily lives may have long-term health consequences; however, society and people have changed in more recent years, and the effects of social determinants may have changed.

It is worth emphasizing that the information on activities such as sports participants and sleeplessness are not collected by survey questions. In the ATUS, experienced interviewers obtained a census of activities across the 24 h (1440 min) of the previous day, activities that may or may not include sports participation and sleeplessness. The duration of activities is known to the minute, unlike surveys that more typically elicit an average effect; further, the recall demanded by the ATUS is only for the previous day, allowing detailed information to be obtained. The time use data are valuable to the US government for learning about time spent on paid and unpaid activities (e.g., childcare, eldercare), the number of hours people work at home and other locations, and how a particular policy might affect people’s behavior. 

Despite these limitations, our study shows strength in the use of minutes to summarize activities during the day. Minutes doing activities can be summarized by the hour, or other time intervals. Minutes doing activities such as sports activities can be summed within other time intervals, if desired. Physical activity is measured precisely by METS across the various activities done during the day. Although not done in this study, minutes would enable a researcher to calculate the activity done a minute before another, minutes of activities within hours, and the minutes between activities. 

## 5. Conclusions

In conclusion, our study of over 20,000 individuals across the US has revealed distinct differences in activity patterns related to demographic characteristics and SDOH. How people spend their day may seem mundane, but differences in daily behavior over time may lead to health disparities. Improving the daily lives of people with SODH should not be overlooked as an approach to primary prevention. As time spent doing one activity takes away from time that can be spent doing another, further research is needed to better understand the optimal distribution across all types of activities (e.g., relaxing, sleeping, vigorous activity), and how best to encourage these activities.

## Figures and Tables

**Figure 1 healthcare-11-01581-f001:**
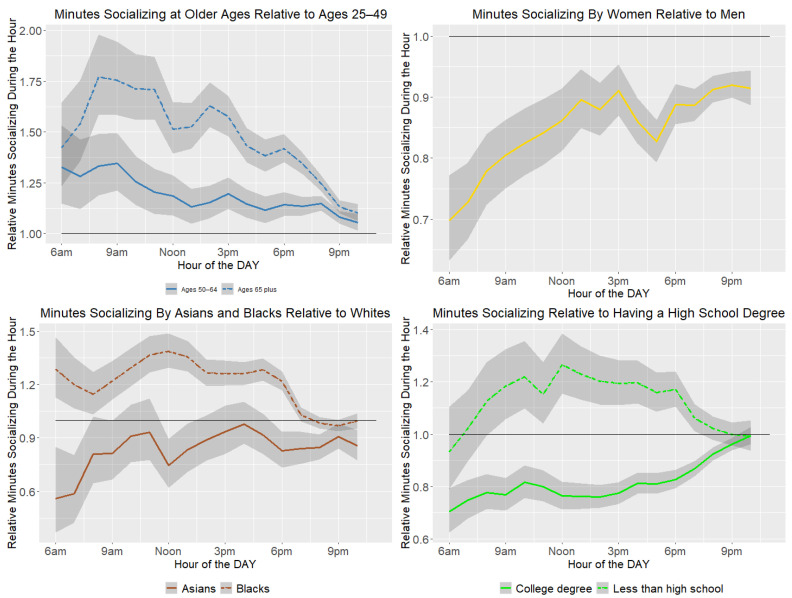
Relative minutes socializing, relaxing, and doing leisure activities during the week by age group, sex, race and education.

**Figure 2 healthcare-11-01581-f002:**
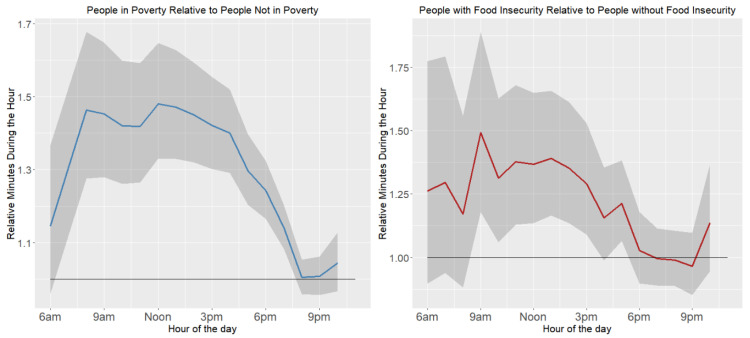
Relative minutes socializing, relaxing, and doing leisure activities during the week by poverty and food insecurity.

**Table 1 healthcare-11-01581-t001:** Descriptive characteristics of the study population.

Variable	Category	Unweighted N	Weighted Percentage
Age group	25–49 years	9229	46.4%
	50–64 years	5757	30.0%
	65 years and older	5340	23.6%
Sex	Males	9219	49.1%
	Females	11,518	50.9%
Race	White	16,209	80.9%
	Black	3566	14.1%
	Asian	962	5.1%
Education	No degree	1227	6.5%
	High school degree	11,081	55.7%
	College degree	8370	37.8%
Weekday or weekend	Weekday	10,340	50.1
	Weekend	10,397	49.9
Living in poverty	No	13,861	70.7%
	Yes	6378	29.3%
Food insecurity	No	19,634	95.6%
	Yes	1013	4.4%

**Table 2 healthcare-11-01581-t002:** Relative minutes of moderate and vigorous activity and of sports activity during the previous day by study characteristics.

Variable	Categories	Moderate and Vigorous Activity		Sports Activity	
		Model 1	Model 2	Model 1	Model 2
Age group	25–49 years	1	1	1	1
	50–64 years	0.93 (0.89, 0.98), 0.01	0.93 (0.89, 0.98), 0.010	0.87 (0.79, 0.96), <0.001	0.86 (0.77, 0.94), <0.001
	65 years and older	0.76 (0.72, 0.80), <0.001	0.76 (0.72, 0.80), <0.001	0.77 (0.69, 0.86), <0.001	0.78 (0.70, 0.87), <0.001
Sex	Male	1	1	1	1
	Female	0.71 (0.69, 0.74), <0.001	0.72 (0.69, 0.75), <0.001	0.59 (0.55, 0.65), <0.001	0.61 (0.56, 0.66), <0.001
Race	White	1	1	1	1
	Black	0.58 (0.54, 0.62), <0.001	0.59 (0.55, 0.63), <0.001	0.61 (0.54, 0.70), <0.001	0.65 (0.56, 0.74), <0.001
	Asian	0.76 (0.68, 0.84), <0.001	0.76 (0.68, 0.85), <0.001	0.94 (0.79, 1.12), 0.53	0.96 (0.80, 1.14), 0.620
Education	No degree	1.02 (0.93, 1.11), 0.73	1.07 (0.97, 1.17), 0.170	1.11 (0.91, 1.35), 0.3	1.25 (1.01, 1.52), 0.040
	High school degree	1	1	1	1
	College degree	0.85 (0.81, 0.88), <0.001	0.82 (0.78, 0.86), <0.001	1.60 (1.47, 1.74), <0.001	1.47 (1.34, 1.6), <0.001
Days	Weekend	1.09 (1.05, 1.14), <0.001	1.10 (1.05, 1.14), <0.001	1.43 (1.31, 1.55), <0.001	1.44 (1.32, 1.56), <0.001
	Weekday	1	1	1	1
Poverty	Yes		0.91 (0.86, 0.95), <0.001		0.71 (0.64, 0.80), <0.001
	No		1		1
Food Insecurity	Yes		0.84 (0.75, 0.93), <0.001		0.76 (0.58, 0.97), 0.040
	No		1		1

Odds ratio (95% confidence interval) and *p*-value are reported. “1” indicates the reference. Pairs of regression models were fit for both moderate and vigorous activities and for sports activities. Model 1 did not include indicators for poverty and food insecurity. Model 2 did include these two covariates.

**Table 3 healthcare-11-01581-t003:** Relative risks of reporting sleeplessness in the previous day by participant characteristics with 95% confidence intervals.

Variable	Category	Model 1	Model 2
Sex	Male	1	1
	Female	1.16 (1.03, 1.31), 0.015	1.15 (1.02, 1.3), 0.030
Age group	25–49 years	1	1
	50–64 years	1.13 (0.97, 1.30), 0.106	1.13 (0.98, 1.31), 0.09
	65 years and older	1.18 (1.02, 1.37), 0.025	1.21 (1.04, 1.4), 0.010
Race	White	1	1
	Black	0.92 (0.78, 1.08), 0.300	0.85 (0.72, 1), 0.060
	Asian	0.50 (0.32, 0.74), 0.001	0.50 (0.32, 0.74), <0.001
Education	No degree	1.34 (1.07, 1.66), 0.009	1.26 (1.00, 1.58), 0.040
	High school degree	1	1
	College degree	0.67 (0.59, 0.77), <0.001	0.73 (0.63, 0.84), <0.001
Days	Weekday	1	1
	Weekend	0.92 (0.82, 1.04), 0.189	1.15 (1.02, 1.3), 0.030
Poverty	Yes		1.21 (1.06, 1.39), 0.010
	No		1
Food insecurity	Yes		1.44 (1.13, 1.82), <0.001
	No		1

Odds ratio (95% confidence interval) and *p*-value are reported. “1” indicates the reference. Pairs of regression models were fit for both moderate and vigorous activities and for sports activities. Model 1 did not include indicators for poverty and food insecurity. Model 2 did include these two covariates.

## Data Availability

The data are freely available with consent at https://www.atusdata.org (accessed on 18 May 2021).

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
