# Peer review of "Socialization, Relaxation, and Leisure across the Day by Social Determinants of Health: Results from the American Time Use Survey, 2014–2016"

_healthcare, 2023, doi:10.3390/healthcare11111581_

Round 1

Reviewer 1 Report

Thank you for the opportunity to review this interesting paper. Based on the American Time Use Survey Datasets(2014-2016), this paper explored the characteristics of individual time use and its influencing factors. The authors need to pay attention to the following problems:

1. The "social determinants of health" in the title might be the theoretical framework of the paper, but the authors had not fully discussed the influencing mechanism of these determinants;

2. The contents of literature review were insufficient, and no specific research hypotheses were proposed, resulting in unclear logic for empirical analysis;

3. The selection of influencing factors of daily activities lacks theoretical basis.

In summary, although the topic of this manuscript is attractive, the paper requires a Major Revision to beef up its theoretical framing, methodology and explanation of results etc.

Reviewer 2 Report

Dear authors, I have reviewed your manuscript and I would like to send you a series of comments in this regard.

-The introduction is too concise, it does not describe the reality that is being studied well, I think that the number of citations could be increased, that they be more current and that they better define the topic under study.

- The aim of the study is not clear.

-In the study population, what criteria were followed to determine that a survey was of poor quality?

- I do not know if these results can be extrapolated and are representative of the population to which they refer. Could you clarify this aspect for me?

-How could you link these results with those of health? I find interesting that possible connection that I did not read in the article in which it is only described.

Thank you so much

Round 2

Reviewer 1 Report

The authors have revised the manuscript according to the comments. The present article basically met the publication requirements.

Author Response

Thank you for your review of our manuscript.  We greatly value your input.  We have made some modifications to the Introduction in response to comments from Reviewer #2 as well as minor changes to the Methods.

Reviewer 2 Report

Dear authors. I thank you for the work you have done but I would like to send you a series of comments/suggestions for improvement.

- From my point of view the introduction does not clarify the subject of study. I consider important an effort in its writing that facilitates its understanding.

- The data you handle is from the year 2014-2016. This raises many doubts about the results. Society, in recent times, has changed substantially and with it people, their situations and I understand that social determinants have also changed. Although its results are comprehensive and well described...can they be extrapolated to the current population? What applicable knowledge does your research bring?

- I consider that the objective can be improved

- I have another question, why was it limited to people 25 years of age or older when the survey collects data from 15 years of age?

- what do they mean when they say people with poor quality data?

Thank you so much
